# Empowering and Educating Parents to Implement a Home Intervention: Effects on Preschool Children’s Engagement in Hands-on Constructive Play

**DOI:** 10.3390/bs14030247

**Published:** 2024-03-19

**Authors:** Michelle Boulanger Thompson, Yaoying Xu, Chin-Chih Chen, Kathleen Rudasill

**Affiliations:** 1School of Rehabilitation Sciences, College of Health Sciences, Old Dominion University, Norfolk, VA 23529, USA; mlthomps@odu.edu; 2School of Education, Department of Counseling and Special Education, Virginia Commonwealth University, Monroe Park Campus, Richmond, VA 23284, USA; ccchen@vcu.edu (C.-C.C.); kmrudasill@vcu.edu (K.R.)

**Keywords:** early childhood, constructive play, parent-implemented intervention, home setting

## Abstract

Constructive play is a creative process-oriented activity that promotes children’s engaged learning through building and designing with materials. This study investigated a parent-implemented intervention to promote active engagement in constructive play for preschool-aged children at risk for developmental delay. This study utilized a single-subject multiple-baseline across-participants design with four participants. Visual analysis of the data identified a functional relation between the temporal, physical, and social–emotional environmental support provided by the parents and the children’s active engagement in constructive play. Parents reported the intervention as meaningful to their lives, indicating strong social validity. These findings highlight the importance of centering and working with parents in their home environment and provide evidence that empowering parents to provide support and minimize barriers facilitates children’s active engagement in constructive play.

## 1. Introduction

Play is universally recognized as a cornerstone of childhood, foundational for learning and happiness [1,2,3,4]. It is through play that children learn about themselves, the physical world, and other people. Play is also the mechanism through which children explore and practice new skills, learn to adaptively respond to their environment, and cultivate a positive self-concept [5,6,7]. Piaget and Huizinga defined play as an enjoyable intrinsically motivated interaction with play materials or other people that does not serve to fulfill basic needs or external goals [2,8]. Constructive play is a creative process-oriented activity that involves hands-on manipulation of open-ended materials to create, design, and build. This type of play is not only fun but is crucial for learning and development [9,10,11]. Through self-directed exploration and creative construction with objects, children investigate, discover, and learn about their world [12].


**Purpose of this Study**


Today, a growing number of young children are not prepared to enter school due to delays in social–emotional readiness that impact their active engagement in learning, a concern that has been amplified by the global COVID-19 pandemic [13]. In 2019, the American Academy of Pediatrics reported a growing concern that changes in today’s lifestyle detract from child-led engaged play at home and contribute to reduced school readiness [14]. These differences refer to today’s hurried lifestyle, family structure, expanded emphasis on enrichment and academic activities, increased electronic screen time, and reduced free play [14,15,16,17].

In the United States, over 40% of young children do not demonstrate the academic or social–emotional readiness they need to be successful in kindergarten [18]. Of particular concern is the number of young children who are “at-risk” for poor social–emotional development due to poverty, trauma and toxic stressors, and unidentified disabilities. These children have reduced opportunities to develop social–emotional readiness through play at home than their typically developing or more affluent peers [19,20,21,22]. The skills gap increases for children living in poverty, with 52% (compared to 25% of children from moderate- to high-income households) not demonstrating the physical well-being, self-regulation, self-management, social–emotional, language, and/or cognitive skills needed for school [14,22]. Toxic stressors and trauma include physical or emotional abuse, chronic neglect, caregiver substance abuse or mental illness, exposure to violence, and economic hardship [14]. It is important to keep in mind that children who experience developmental delays or disabilities are at higher risk for future academic, mental health, and behavioral difficulties [14,15,23,24].

To ameliorate the effects of toxic stress, poverty, and disability, the American Academy of Pediatrics (AAP) recommends teaching children resilience through play. The AAP also recommends play in the family context as a best practice for promoting healthy child development and social–emotional well-being [14,16]. Given that play and child development are connected, and the impact of the home environment is critical for child development and school readiness, this study explores a parent-implemented intervention to promote active engagement in constructive play for preschool-aged children who are at-risk for developmental, social–emotional, or behavioral disability. Parent education about the importance of play and the physical, temporal, and social–emotional home environment enables parents to provide support and minimize barriers, facilitating their child’s active engagement in constructive play.

The essence of constructive play is the joy of creating with and exploring play materials. It sparks imagination, deepens engagement, and can lead to pretend play. **Children spend a significant portion of their time engaged in constructive play. Studies show it occupies 40% of a three-and-a-half-year-old’s play and increases to 50% for four- to six-year-olds [10,25,26,27]**. Constructive play also serves as a foundation for school readiness skills. Although best known for its correlation with early math development, constructive play also builds growth in preacademic literacy and fosters essential skills such as problem solving, cognitive flexibility, and emotional readiness and regulation [12,28,29,30].

The interrelationships between children’s active engagement, play activities, and school readiness are well-documented in the literature [14,16]. Additionally, there is much literature supporting the practice of working with parents to implement interventions to improve language, communication, social–emotional, behavioral, and other developmental skills in young children [31,32,33,34,35]. However, the impact of preparing parents to facilitate children’s play has far less presence in the literature [36]. Therefore, this study investigates the effects of a parent-implemented intervention on active engagement in constructive play. The following research question guides this study:

Do parent-implemented environmental support strategies improve the child’s active engagement in constructive play in the home?


**Importance of Play for School-Readiness**


Children’s play is a primary vehicle for learning in early childhood and is related to acquiring both preacademic and social–emotional school-readiness skills [7,14,16]. It also provides young children the practice and opportunity to respond to their environment adaptively, building emotional readiness and coping skills foundational for active attention and engagement [37,38]. One of the most common types of play in the preschool years is constructive play, an active, hands-on type of play where children build and combine objects to experiment and enjoy the creative process of construction [9,39]. A critical component of social–emotional learning is self-management, a skill that enables children to regulate their emotions and behaviors and to persevere with challenging tasks [40]. In children’s play, active engagement in the play activity demonstrates that self-regulation and self-management are being engaged and practiced [40,41]. 


**Constructive Play’s Contribution to Development**


Constructive play emerges from functional play with objects at around age two and blends with imaginary play around ages four or five, becoming more complex and creative over time [7,10,42]. Through hands-on manipulation of play materials, children develop spatial literacy, cognitive flexibility, and mathematical classification knowledge such as color, size, shape, texture, and sequencing [43]. This creative exploration lays the foundation for future success in academic subjects like math, architecture, and engineering [44,45,46]. Additionally, research shows that constructive play in early childhood correlates with literacy and language development and that hands-on play with nonelectronic play materials is associated with improved quality and quantity of language growth [47,48]. 

The literature identifies several genres of constructive play, including tinkering, loose-parts, engineering, construction, and makerspace play activities all which encourage experimentation, creativity, problem solving, critical thinking, and a sense of agency [7,10,30,49,50,51,52,53,54,55]. In other words, constructive play allows children to express and explore their ideas. But this creative engagement happens in context. Nicholson’s 1972 theory of loose parts emphasizes the importance of the environment in fostering creativity, engagement, and discovery in constructive play, and how the environment mediates how engaged and creative children become during constructive play [55]. However, due to its overlap with functional and imaginary play, constructive play is not as well-represented in the literature as research focuses more on functional, pretend, or social play skills.


**Importance of the Home Environment**


Pioneering research by Bronfenbrenner, Bandura, Piaget, and Vygotsky underscores the vital role the environment has on children’s development, including play [2,56,57,58]. Play provides practice and opportunity, empowering children to build the emotional readiness and coping skills necessary for active engagement, exploration, and creativity [6,30,37,38,59,60,61,62]. Research shows that the natural home environment contributes to developing play, playfulness, and emotional-regulation skills, impacting the child’s active engagement, attention, and participation [6,62,63,64,65]. 

Throughout the literature, we find environmental practices that support the development, play, and learning of young children, including those with or at-risk for disabilities. These fall into several distinct categories that address the child’s interaction with their physical environment (space, materials, and sensory input), their social–cultural environment (family members and friends), and their temporal environment (time and routines). By mindfully curating supports in their home environment, parents can nurture and facilitate their child’s learning and development, health and safety, and engagement in play [66,67,68]. 


**Importance of Parent-Implemented Interventions**


Parents are the primary influence on their young children’s learning and development in the natural home environment. As noted throughout the literature, parent-implemented interventions are a successful, evidence-based method of effecting change for children and families [31,33,34,35,69,70,71,72]. Benefits of parents providing the intervention include increased parental and family capacity to support the learning and development of their children, reduced parental stress, improved parental responsiveness to their child’s needs, and acquisition of the ability to practice and generalize the intervention across natural environments [73,74]. Unfortunately, research on parent-implemented interventions for preschool-age children’s play in home and community settings is sparse [75]. The preponderance of research has focused on parent-implemented language, communication, and behavioral strategies. Relatively little research has focused on interventions to facilitate play [36]. 

## 2. Methodology


**Design**


We conducted a pilot study to test the feasibility of the instructional materials, intervention procedures, measurement and data collection system, and communication modalities, and to affirm the social meaningfulness of the intervention for the parents and children. For the pilot study, we trialed the proposed research study materials and processes using two parent–child dyads as participants. A nonconcurrent A–B design was utilized for this pilot study which helped inform this study utilizing a multiple-baseline across-participants design. The pilot study affirmed the social validity of this study and provided efficacy of the parent education materials and process.

We applied a single-subject multiple-baseline across-participants design to examine the effects of the parent-implemented intervention on their child’s active engagement in constructive play. Single-subject research design (SSD) provides experimental rigor to test a novel intervention with only a few participants and allows for the individualization and accommodation necessary for research in nonclinical naturalistic settings [76,77]. This design often functions as a preliminary type of research to establish a base of knowledge about the efficacy of the intervention before testing with larger groups [76,77]. Given the limited research on constructive play and, more specifically, on parent-implemented play interventions in home settings, we used a multiple-baseline across-participants design as it is a research design of choice in the social sciences used to evaluate an intervention’s effectiveness to improve behavior [76,78,79,80]. This design provides experimental control as the concurrent baseline phases are followed by staggered intervention conditions across all participants to validate documented effects. This design was selected also because it provides ethical considerations of participant needs, over the ABAB design, as there is no withdrawal or reversal of the treatment. 

Child outcome data were collected on the child participants’ active engagement in constructive play during baseline, intervention, and maintenance phases. The threat to the reliability was minimized through the establishment of interobserver agreement (IOA), which ensured that the variations and inconsistencies in observation were minimized, individual observer biases were limited, and that the targeted behavioral outcome was well-defined [77,81]. For this study, a recent doctoral graduate in special education with experience working in early childhood special education served as the research assistant to review and score play recordings using a researcher-created data collection tool. As recommended by Ayers and Ledford and Ledford et al., IOA was assessed for at least 33% of sessions, and at least an 80% agreement level was achieved between the researcher and the research assistant across all conditions [79,82]. The point-by-point method, recommended by Gast and Ledford and Kazdin, was used to calculate the mean IOA percentage by dividing agreements by the sum of agreements plus disagreements, then multiplying by 100 [77,78]: agreement/agreement + disagreement × 100 = % agreement

To ensure fidelity of implementation of the intervention in the home environment, we documented observed parent practices of the intervention protocol for each submitted play recording and coded these observations using a researcher-created fidelity of intervention sheet. Parents were also asked to complete a survey after each recorded play session to guide their self-reflection and to document their fidelity of implementing the intervention. Surveys were provided electronically using a Google Form survey through a secure university server. 

Social validity was assessed to discern the meaningfulness and social impact of the intervention in the lives of the participant parents and children [76]. To ensure that the intended outcome and the intervention process were relevant to their family, we met with each parent prior to the intervention phase to discuss the family’s unique culture, needs, and concerns. This preliminary interview allowed us to provide a more individualized and meaningful intervention experience that was mindful of each family’s unique culture as well as their child’s developmental needs and preferences. The social validity of this study was then assessed using a survey completed by the parent with input from the child.


**Setting and Participants**


This study was conducted in an urban mid-size mid-Atlantic city. The setting for data collection during play activities was the indoor home environment of each child and their participating parent. Children and parents were recruited from local public and private community-based preschool centers serving low-income and at-risk children. The first author emailed preschool directors to inform them about the purpose of this study and included a recruitment flyer to share with their teachers and prospective parents. Children selected to participate met inclusion and exclusion criteria (listed below), were identified as at-risk for disability by their preschool teacher or parent, and demonstrated developmental, social–emotional, or behavioral difficulties that affected their play at school or home. For purposes of this study, “at-risk” is defined as having a diagnosed or suspected disability in the categories of developmental delay, autism spectrum disorder, or attention deficit disorder, or being at-risk for a disability due to trauma or poverty. Five children were accepted and began this study, but only four completed the baseline data collection phase. 


**Inclusion and Exclusion Criteria**


Inclusion and exclusion criteria for participating children and parents include the following:Child participants are 4-years-old for the duration of this study;Child participants attend a community-based inclusive preschool (i.e., Head Start);Teachers and/or parents express concern with the child’s engagement in play;Child is considered at risk for a developmental delay. Risk categories include one or more of the following:Suspected or documented disability of developmental delay, autism spectrum disorder, or attention deficit disorder;A history of economic hardship or insecurity (poverty);Have experienced toxic stress (history of trauma, exposure to abuse or violence, caregiver substance abuse, caregiver mental health issues, physical or emotional abuse, or chronic neglect);Parent participants are the custodial guardian with whom the child resides four or more days weekly;Parent participant speaks English conversationally with the researcher and participates in this home study;Child does not have an orthopedic impairment that affects the upper extremities, such as cerebral palsy;Child is not enrolled in a self-contained special education classroom in a public school.


**Antonio**


Antonio (All names are pseudonyms to protect the confidentiality of the participants) is a 4-year-old boy who lives in an apartment with his parents and baby sister. He is considered at risk for developmental delay due to the risk factors of poverty and economic hardship, and a current delay in speech and language. His mother reports he previously received early intervention services, and she continues to worry and have significant concerns about his play and overall general development. He attends an inclusive Head Start preschool close to his home. His mother is concerned about his ability to ask for help and how he is easily frustrated when he makes what he perceives as a mistake, even when he is playing. 


**Kiki**


Kiki (All names are pseudonyms to protect the confidentiality of the participants) is a 4-year-old girl living in a house with her mother, grandmother, twin brother, and two older brothers close in age, and stays with her father on weekends. She is considered at risk for developmental delay due to the risk factors of poverty and economic hardship, history of trauma in the family, and a suspected diagnosis of autism. Her mother reports that although Kiki received early intervention services to address her general development and her speech, currently she does not receive services even though both her mother and her teacher have concerns with her play, social, and communication skills. Kiki attends an inclusive Head Start preschool close to her home. 


**Mateo**


Mateo (All names are pseudonyms to protect the confidentiality of the participants) is a 4-year-old boy who lives in a suburban neighborhood with his parents. His parents both work full time; his mother is a public-school teacher and his father is self-employed. His mother is also a part-time student pursuing a doctoral degree. His grandparents also live nearby and are involved. His home is bilingual English and Spanish. His father recently immigrated and purposefully only speaks Spanish to Mateo while his mother speaks to him in both Spanish and English. Mateo is considered at risk due to a genetic diagnosis of NSUN2 which results in a global developmental delay, ADHD, autism, a severe speech delay, and cerebral palsy that affects his trunk and lower extremities. 


**Jayce**


Jayce (All names are pseudonyms to protect the confidentiality of the participants) is a 4-year-old boy who lives with his mother in public housing. However, due to poor maintenance and unhealthy living conditions of the apartment, Jayce and his mother were temporarily staying with their extended family, a 40 min drive from his inclusive community Head Start preschool. He is at risk for a developmental delay due to poverty and economic hardship, toxic stress of current living conditions, increased use of electronic screen time, and a suspected diagnosis of ADHD. Jayce’s mother and teacher are concerned about his play with toys and with other children. Jayce’s mother reports that his strength is his ability to play on his own independently. 


**Materials**


Materials for baseline play sessions were chosen by the parent and child. Play materials introduced during parent instruction in the intervention included toys, household objects, familial or cultural items, arts and craft materials, sensory mediums, or items from nature. The first author brought novel construction play materials to each child to ensure families had access to materials, choices, and novel items to offer their child. These provided play materials were identical for each participant and included small domino-size colorful wooden blocks, multi-colored craft popsicle sticks, and homemade playdough. These materials supplemented the toys and play materials already in the child’s home. Additional materials provided include the use of an Apple iPad (5th generation) for videorecording, a digital copy of the intervention PowerPoint slides, and access to a secure google drive to upload recordings and google forms for parents to report their fidelity of treatment implementation.


**Children’s Active Engagement in Constructive Play**


The primary dependent variable (DV) in this study is operationally defined as the child’s active engagement in constructive play in the home environment, modified from the definition developed by DiCarlo et al. [37]. For purposes of this study, constructive play is defined as any hands-on activity with more than two toys, materials, or items from the household or from nature that the child combines to create, build, or construct [9,39,42]. This definition of constructive play aligns with the general description of play as a pleasurable and enjoyable interaction with toys, objects, or other people that is intrinsically motivated and does not serve to meet a basic need or achieve an externally defined goal [2,8]. It also aligns with play as creative, meaningful, joyful, and engaging for the child, as evidenced by emotionally regulated active engagement. Active engagement in play requires the child’s affective involvement and interest so that play is joyful [83]. 

In this study, active engagement in constructive play is evidenced by two types of observable behaviors. First is the demonstration of interest in constructive play noted by the child’s hands-on engagement with play materials to build, construct, or combine to create structures or designs. Alternatively, constructive play can be demonstrated socially by the child showing, telling, or asking the parent about their construction. The second type of behavior indicates that the child is in an emotionally regulated state, optimal for exploration, creativity, and engagement. For this study, expressions of pleasure, happiness, or playfulness are demonstrated by the child smiling or laughing. Positive emotional affect can also be demonstrated by the lack of emotional dysregulation such as lack of crying, fussing, yelling, or the expression of negative words such as “I hate this”, or “I don’t want to”.

The active engagement in constructive play was measured using partial interval recording every 20 s over a 5 min period to document evidence of the child’s engagement in constructive play. A score of 1 point was awarded when the child demonstrated an observable hands-on interaction with play materials or social sharing of the play materials or process with their parent, along with the demonstration of being emotionally regulated. Measurement of the dependent variable was documented and a total score per play session was calculated and graphed. This score was determined reliable by calculating interobserver agreement (IOA) between the researcher and researcher assistant observing video-recorded play sessions during baseline, intervention, and maintenance phases.


**Parent Play Facilitation**


The independent variable (IV) in this study is operationally defined as the parent-implemented physical, temporal, and social–emotional supports and the reduction in barriers the parent puts into practice at home to facilitate engaged constructive play for their child. The first author met with each parent virtually for 60 min to explain the definition of play, benefits of constructive play, how to implement the intervention, provide visual examples, help parents set personal goals, and facilitate self-reflection about their child’s current play and their current environmental supports. 


**Procedures**


To facilitate children’s active engagement, parents modified their home’s physical, temporal, and social–emotional environment by providing supports and removing barriers. Aligning with the findings from DiCarlo et al., parents were guided to modify the physical environment by turning off distractions such as electronic screens, provide limited familiar and novel choices of play materials, and ensured the physical play space as safe and comfortable to meet their child’s unique sensitivities [37]. As recommended by research by Kiewra et al. and Knox, parents supported the temporal environment by ensuring their child’s basic needs were met and time for play was in their child’s daily routine and schedule; further communicating play as valued by the family [84,85,86,87]. Finally, to modify the social–emotional environment, parents supported child-led play by being present, playful, and available [88], being emotionally responsive to their child frustrations, asking open-ended questions, and inviting their child to socially share their creations [84]. 


**
*Baseline Phase*
**


The baseline phase for all four participants began the same day, as it was essential that baseline data were collected concurrently to strengthen the experimental control of this study’s design [79]. The first author instructed parents in the video recording protocol and asked parents to record their child’s play in their home saying, “Please video-record your child playing for 10 min, as they typically play”. No other guidance was provided as the purpose was to record the child’s play without a prompt to the parent to alter the home’s physical, temporal, or social–emotional environment. The first author used a researcher-created data collection sheet to document the child’s active engagement in constructive play activities, completing the form when viewing the recorded play session then graphing the scores.


**
*Parent Instruction and Transition to Intervention Phase*
**


The transition between baseline and intervention condition occurred spontaneously and sequentially across participants as they submitted baseline data that were stable and trending in a zero-accelerating or decelerating direction. The order of participants moving from baseline to the intervention phase was Antonio, Kiki, Mateo, then Jayce, based on the order they submitted baseline recordings and met criteria to move to the intervention phase. Following the multiple-baseline design principles, readiness of the next participant to move into the intervention phase depended on the stability of the first three data points of the intervention phase of the prior participant [78]. 

When criteria to transition out of baseline were fulfilled, the first author met with each parent virtually over zoom for a 30 min parent interview followed by a separate 60 min scripted PowerPoint presentation to teach parents about how to provide the intervention in their home for their child. To teach parents to facilitate constructive play, the first author presented photos of play materials, shared photos of children engaging in building and designing, described the benefits of play, and discussed strategies to provide supports and reduce barriers in the home’s physical, temporal, and social–emotional environment. 

A portion of this parent instruction included opportunities for parents to set personal goals for themselves related to strategies they learned to promote changes in their home’s temporal, physical, and social–emotional environment. Coaching, modeling, and opportunities for role playing, self-reflection, and performance feedback were provided to the parent during this instructional session, a follow-up 30 min coaching session, and follow-up text conversations. During the active discussion sections of the presentation, parents prioritized goals for themselves in each area. The first author reviewed these goals with parents in their follow-up coaching and text conversations, provided parents a copy of their goals for reference and reflection, and provided a copy of the PowerPoint slides. Once parents were instructed on the intervention protocol, and the first author confirmed their understanding and mastery of the content using a researcher-created quiz, they were instructed to begin the intervention phase. This process of parent instruction was repeated consecutively with the remaining parents. 


**
*Intervention Phase*
**


In the intervention phase of this study, parents implemented the intervention, making changes in their home’s physical, temporal, and social–emotional environment to facilitate their child’s engagement in constructive play. They were instructed to record 10 min play sessions three or more times weekly. We coded the middle 5 min of each 10 min session to ensure stability was established. They were also asked to complete a parent self-assessment fidelity checklist after each recorded play session which functioned as a reminder checklist and a self-rating on their fidelity of implementation. During the intervention condition, the first author texted parents photo examples of constructive play from the PowerPoint slides as well as these self-reflection questions to encourage and facilitate their children’s play: (a) Did I set aside time today for play? (b) Was the play space safe and inviting for my child? (c) Did I offer choices in play materials? (d) Was I emotionally available to my child? And (e) did I ask open-ended questions and offer encouragement?


**
*Maintenance Phase*
**


Maintenance data were solicited 3–10 weeks after the last intervention score was recorded. During this time, parents were encouraged to continue promoting play at home for their child but did not complete daily play fidelity checklists as they had during the intervention phase. Like the baseline period, parents were prompted to record their child playing for 10 min as they typically play and to capture three or more recordings. No text reminders about constructive play were provided during this final maintenance phase and there was no communication with the research team.


**Data Collection and Visual Analysis**


Data were collected using partial interval recording which is considered a preferred method of measuring the occurrence of a behavior [82,88]. Every 20 s over a 5 min period of each play recording, the evidence indicating active engagement in constructive play was marked. One point was assigned for each occurrence of active engagement as operationally defined. Each play recording had a possible score of 15 total points, which were then graphed.

The data were then transcribed onto a visual line graph with the child’s active engagement represented on the *y*-axis and time represented on the *x*-axis. The level, trend, and variability of the dependent variable within and between conditions were analyzed to discern a functional relation between the child’s active engagement in play and the parent-implemented support, and to assess experimental control [76,77,78]. 

## 3. Results


**Visual Analysis of Graphic Data**


When assessing the changes in level between phases, there was a visible positive change in level between the baseline (condition 1) and the intervention (condition 2) for all four participants, and this change was visibly maintained after the intervention phase concluded (condition 3). When visually assessing for trend and stability, Mateo demonstrated accelerating and stable trendlines in the intervention phase while both Mateo and Antonio demonstrated accelerating and stable trendlines in the maintenance phase, further supporting the functional relation between the parent-implemented intervention and the child’s engagement in constructive play (see Figure 1). 


**
*Variables Within and Between Conditions*
**


Analysis of variables within and between the three conditions was utilized to augment line-graphed visual data (see Table 1 and Table 2). Within condition, calculations include the level length, range, mean, median, level absolute change, level relative change, trend strength and direction, and level and trend stability (see Table 1). R-squared (*r*^2^ = 0.00 to 1.00) explains the strength of the relationship between the parent-implemented support and the child’s active engagement and was calculated to augment the visual analysis of the changes in strength and directionality of the trendlines in each condition. Of note, all four participants submitted at least one follow-up maintenance-phase recording 3 to 10 weeks after their final intervention phase submission, providing a true break in conditions between the intervention and maintenance phases.

Between condition, calculations include comparisons between phases in level mean change, level median change, level absolute change, and level relative change. Additionally, the level stability, trend direction and effect, and percent of overlapping (POD) and nonoverlapping (PND) data between all phases were assessed to determine the magnitude of the effect and the impact of the intervention (see Table 2). When assessing level changes between phases, all four participants demonstrated an immediacy of effect when calculating the absolute change between the last data point of the baseline phase and the first data point of the intervention phase. The median difference, mean difference, and relative level change between the baseline and intervention phases also support what we see through visual analysis of the graphic data and suggests a functional relation between the intervention and the observed outcome for all participants. The relatively small change in median, mean, absolute, and relative levels between the intervention and the maintenance conditions further supports the effectiveness of the intervention. The PND for all participants is 100%, demonstrating no overlapping data points between the baseline and the intervention conditions, and a PND of 0%, illustrating full overlap of data scores between the intervention and maintenance phases, supporting the strong magnitude of the effect of the intervention. 


**
*Participant Performance*
**



**Antonio**


Antonio’s mother proceeded to record and upload seven videos daily for the first week. The child’s baseline trend and level were stable, so Antonio’s mother was instructed in the intervention and began the intervention phase of data collection on day 12. Antonio demonstrated an immediacy of effect when looking at the absolute level change between the last data point of 0 in the baseline phase to the first data point of 13 in the intervention phase. This agrees with our finding using visual analysis of the graphic data that the change in level for Antonio between baseline and intervention conditions indicates a functional relation between the parent-implemented intervention and the change in the child’s engagement in constructive play in the home environment. 

Antonio’s engagement in constructive play scores sharply decreased after the first three intervention recordings, but this was primarily due to the parent encouraging and leading her child in dress-up and imaginary play, not constructive play. This parent abruptly stopped submitting recordings and ceased all contact with the researcher (text, email, and phone) a little over two weeks into their intervention phase. It is unclear but this family may have been in crisis as Antonio returned ten weeks (74 days) later to participate in maintenance data collection and to complete the post-study parent survey. After this ten-week pause, Antonio’s parent submitted three play recordings that demonstrate the fidelity of the parent intervention, and the subsequent child outcome was maintained. The effect and generalization of the independent variable, the parent-initiated play intervention, was supported with calculations of the improved level mean, median, absolute, and relative changes. Additionally, 0% percent nonoverlapping data (PND) and 100% percent overlapping data (POD) between the intervention and the maintenance conditions reveals support that comparison of condition levels indicates the intervention as the root cause of the change in child play engagement. Analysis of trendline direction and stability, however, does not support this finding.


**Kiki**


Kiki’s mother submitted six baseline recordings with a baseline trend and level stable at r = 0. The first three data points of Antonio’s intervention phase demonstrated an upward trend of r^2^ = +0.571, so Kiki’s mother was instructed in the intervention and began the intervention phase of her data collection on day 19. There was an immediacy of effect when looking at the absolute change between the last data point of 0 in the baseline phase to the first data point of 13 in the intervention phase. Kiki’s engagement in constructive play scores were high for the first three intervention recordings but then drop, possibly impacted by the environmental stressors of a sick family member and the need to relocate their residence.

There was a consecutive four week (28 day) pause between Kiki’s intervention phase and the collection of follow-up maintenance data. A single play recording was submitted by this participant for the maintenance phase, so trend in the maintenance condition could not be calculated. The single score of 15/15 suggests generalization of the intervention carried forward as evidenced by the improved level mean, level median, and relative change in levels, a stable absolute level change, 0% PND, and 100% POD between the intervention and follow-up conditions. Like Antonio, Kiki’s visual and statistical analysis of the level changes suggests a functional relation between the intervention and the outcome, while analysis of trend and stability is inconclusive. 


**Mateo**


Mateo’s mother submitted nine baseline recordings with low variability scores ranging between 0 and 3 with the baseline trend at r^2^ = 0.01. It should be noted that originally this last data score of 3 was coded as a 2 but recoded as a 3 upon review. The minimally variable nature of this baseline trendline allowed the researcher to determine the readiness of this participant to move into the intervention phase. To further assess readiness to transition Mateo to the intervention phase, the first three data points of the previous participant, Kiki, were calculated at +0.571, indicating a strong upward trendline. Mateo’s parent was then instructed in the intervention and began the intervention phase of data collection on day 27. Mateo demonstrated an immediacy of effect when looking at the absolute change between the last data point of 3 in the baseline phase to the first data point of 11 in the intervention phase. 

After a four week (29 day) pause, Mateo’s parent submitted five play recordings over a 5-week span for the maintenance condition. The effect and generalization of the independent variable, the parent-initiated play intervention, was supported by improved level mean change, level median change, absolute level change, relative level change, 0% PND, and 100% POD between the intervention and the maintenance conditions.


**Jayce**


Jayce’s parent collected only three baseline data points. This was due to a crisis with her housing and her need to relocate with her child to stay with family, leaving minimal time and privacy for her child to play and for her to record. These three baseline recordings demonstrated a baseline trend and level that were stable at 0. After requesting but not receiving additional baseline recordings and confirming the stability of the first three data points in the intervention at 0.563 for the previous participant, Mateo, the researcher elected to move Jayce into the intervention phase. Jayce demonstrated an immediacy of effect in his engagement in constructive play when looking at the absolute change between the last data point of 0 in the baseline phase to the first data point of 7 in the intervention phase. This immediacy of effect may have been lessened by the parent waiting 23 days between parent training and when intervention-phase play recordings were produced. Additionally, this parent reported that her child was struggling to transition back to her home from weekend visitation with his father when the first two intervention phase recordings were produced, possibly contributing to these initial lower intervention-phase data points. 

After a three week (20 day) pause in communication with the researcher, Jayce’s parent submitted one play recording that was coded at 15/15, suggesting that the effects of the intervention continued. Although the trend in the maintenance condition could not be calculated using a single data point, the generalization of the intervention is supported by the improved level mean, level median, and relative change in levels, a stable absolute level change, PND, and POD between the intervention and the maintenance conditions.


**Interobserver Agreement Results**


Recordings were randomly selected across participants and phases for the research assistant to observe and code. The research assistant who helped with IOA for this study has over 25 years’ experience working with young children with developmental disabilities and recently graduated with a Ph.D. in special education. Prior to collecting IOA, she was trained to observe and discern constructive play and instructed to code using the data collection sheet with specific behaviors operationally defined. Interobserver agreement was established across 15 time slots for three 5 min recordings at 100% before coding additional recordings independently. IOA was determined to be acceptable, above 80% per Horner et al., with 87–100% IOA in the baseline phase, 87–100% IOA in the intervention phase, and 100% IOA in the maintenance phase across all participants [76,77].


**Fidelity of Intervention**


Treatment fidelity is one of the quality indicators of single-subject design research [76,78]. In this study, the intervention is the strategies the parent implements to provide supports and reduce barriers in the home’s temporal, physical, and social–emotional environment. 

Parents were instructed in the purpose and strategies of this intervention through dialogue with the researcher and presentation of PowerPoint slides. After initial instruction, parents’ knowledge was assessed using an online quiz and with follow-up discussion with the researcher. All parent participants were able to demonstrate an 88% pass rate on their own, with errors then discussed and retaught by the researcher to achieve an overall 100% understanding of the intervention content.

During the intervention phase, parents were asked to provide a self-assessment of how they implemented each component of the intervention. This parent self-assessment was presented in the form of a checklist on a secure Google Forms platform that they could access using the provided iPad, their cell phone, or their home computer. Parents were also provided the opportunity to describe how successful they felt in supporting their child’s constructive play each session, either by providing a narrative within the google form checklist or by texting their feedback. Antonio’s mother completed 6/6, Kiki’s mother completed 1/6, Mateo’s mother completed 9/9, and Jayce’ mother completed 0/4 of the requested online self-assessment checklists. Additionally, all parents shared some form of self-assessment as they all independently elected to text the researcher after each play session to express their observations about their implementation process, to share changes in their child’s play, and to confirm they had uploaded a new play video. 

The researcher viewed and coded each video for fidelity of intervention implementation by the parents using the fidelity of intervention coding sheet. Antonio’s, Kiki’s, and Mateo’s parents demonstrated high fidelity of implementing the temporal, physical environment, and social–emotional supports of the intervention, including asking open-ended questions, making open-ended comments, being emotionally available to their children, prioritizing play and creating time for play in their daily schedules, and creating a child-friendly nondistracting play area with choices of play materials. It should be noted that many of the recordings that received lower scores in the intervention phase had good fidelity of implementations, but scores were reduced when the play pivoted from constructive play to imaginary social play between child and parent.


**Social Validity**


Parents reported the intervention’s value in their post-study survey, texts, and coaching calls. They described their participation in this study as meaningful with a positive impact on their daily lives.

“I understand now how important play time is. I understand how important it is to set a scene and participate and communicate when my children are playing”.

“I really enjoyed this experience and learning more ways to help my child learn and explore”.

“Thank you for encouraging us to find more ways to inspire him to be more constructive and creative. Also, for me as the parent to uplift him more in his process”. 

“At first he used to just play with cars and trucks and now he’s able to make better ideas on what he wants to play with and what play he wants to do that day, and if he needs help”.

“Me focusing on helping her play is also helping her learn self-advocacy through making choices”.

“Turning off TV really does make a big difference in how he plays and what he plays with”.

“This has taught not only him but myself as a parent to help create and enhance my child’s ability to construct, play, and enjoy doing it and learning. Also, for me to assist when needed and ways to boost his esteem when having difficulty playing”.


**Generalization**


The intervention shows promising signs of generalization, meaning children used their newfound play skills in new settings and with unfamiliar people. Antonio’s mother shared that her son now includes his father and sister in his play, even though his father had not received training in the intervention, saying, “He’s more able to include his father and sister in his play and talk about his plan of action as he plays as well”. 

Similarly, Mateo carried his new play skills to preschool. His preschool teacher, impressed by his new creative construction skills, texted his mother, “Your little builder- so creative and so cute!”. 

## 4. Discussion

The results of this study suggest a functional relation between the parent-implemented play intervention and increased engagement in constructive play using nonelectronic play materials in the home environment. This is demonstrated through visual analysis showing an immediate and significant improvement in play engagement upon introducing the intervention. Additionally, anecdotal evidence suggests that this parent-implemented intervention may generalize across different people and settings. These findings highlight the importance of centering, supporting, and empowering parents to directly contribute to their children’s development through constructive play.

This study’s social validity is strongly supported by the parents’ appreciation that the intervention not only enhanced their children’s play skills, but also strengthened their parent–child relationships and improved daily home life. Future research, professional development, and policy should prioritize initiatives that support, center, and empower parents in their children’s development and learning through constructive play. 

These results honor and affirm the fundamental role of parents in nurturing play in children within the cultural context of the family. We aligned the intervention with the Division on Early Childhood’s recommended practices that guide us to consider the environment, capacity-building, and the provision of family-centered care for young children [66]. Our intervention empowers parents by embedding opportunities for participation, choice making, and self-reflection. We promote capacity building and autonomy for parents by encouraging parents to plan, make decisions, and set personal learning goals that reflect their family’s unique needs and their child’s personality and interests. We address the DEC recommended practice of recognizing the value of an accessible and safe natural environment as optimal for young children’s learning and development. We accomplish this by guiding parents to tailor the physical, temporal, and social–emotional aspects of their home environment to best meet their child’s individual needs, which aligns with Bal and Trainor’s quality indicators of culturally responsive research that recognize the importance of meaningful and culturally responsive research [89].


**Meaningfulness of this Research**


In addition to increasing children’s engagement in constructive play, the intervention provided meaningful benefits for families. Parents reported that they learned to support their children’s individual needs and to recognize the crucial role of play in their children’s development and emotional well-being. They also reported that their children demonstrated improved confidence, communication, self-advocacy, persistence, and social relationships. In the words of Jayce’s mother, “This study has really taught us both to do things more together”. Kiki’s mother explained, “Me focusing on helping her play is also helping her learn self-advocacy through making choices”. Antonio’s mother described the benefit of her son learning emotional regulation through the changes she made at home, sharing, “At first he used to shut down and not able to explain his frustration or difficulty but now he can in a simple detailed way without getting upset or shutting down”. This study was sustainable and cost-effective for families. Parents were able to implement the 10 min play session interventions using readily available materials in the home. Measuring the practical significance of this intervention was not in the scope of this study, but we envision this intervention as beneficial to a wider range of individuals, including parents, early childhood teachers, and childcare professionals. 


**Limitations**


This study acknowledges several limitations, a small sample size, fidelity of intervention implementation, and threats to experimental control due to the global COVID-19 pandemic and variability of the data. 

Single-subject multiple-baseline across-participants design research relies on multiple replications between phases with at least three participants experiencing concurrent baseline conditions [81]. In this study Antonio, Kiki, and Mateo began data collection in the baseline condition concurrently, which meets the WWC criteria. The fourth participant, Jayce, began later. Additionally, Antonio, Kiki, and Mateo each produced five or more data points in both the baseline and intervention conditions, meeting What Works Clearinghouse single-case standards “without reservations” [82]. Jayce only produced three data points in the baseline and intervention conditions, meeting WWC standards “with reservations”. A larger sample size would accommodate inconsistent participation and potential participant attrition. 

Ensuring consistent fidelity of intervention implementation by parents was challenging. The intervention involved multiple components of simultaneously assessing their child’s physical, temporal, and social–emotional environment then meeting their child’s ongoing needs by providing support and reducing barriers to facilitate increased engagement in hands-on constructive play. While efforts were made to ensure consistent application of the intervention, potential variations in how parents implemented it could affect the results. Only two of the four parents consistently self-reported their fidelity of implementation. All intervention videos were coded for fidelity of parents implementing the intervention; however, this was limited to what was captured on the recordings. A more optimal camera angle, clearer audio, and more frequent self-reports from parents could have provided a more complete picture and improved fidelity of the intervention.

The quality indicators for single-subject research guided this study and helped us control for internal validity threats [76]. Threats to experimental control were minimized by the multiple-baseline study design, a shorter duration study limiting natural maturation in young children, and a thorough description of external events, particularly the global COVID-19 pandemic which spanned the duration of this study. Despite efforts to control for internal validity threats, some challenges remained. Visual analysis of data was promising, but data trends and variability from three participants during the intervention phase did not definitively demonstrate experimental control [90]. We noted Antonio, Kiki, and Mateo’s mothers intermittently glancing at the recording device, which may indicate their self-consciousness about performing on camera and influencing their natural interactions with their children. This is known as the Hawthorn Effect, which is considered a threat to experimental control. **Variability in parental presence is also a threat.** It is not clear why Jayce’s mother was not visibly or audibly present during any of his recordings, limiting her ability to be supportive during her son’s play, and greatly impacting the observed intervention fidelity score.


**Implications for Research**


This study establishes initial evidence that with training, coaching, and support, parents are capable of increasing their children’s hands-on constructive play at home. The limitations described above suggest possible modifications for future research. To address the limitation of treatment fidelity, in vivo coaching sessions in the family’s home would provide a glimpse into how the parent sets up for play, how they implement physical, temporal, and social–emotional supports for their child, and allow the researcher to provide parent coaching as the need arises.

Training teachers, childcare professionals, and related service providers to provide parent education and coaching on the intervention would be a good expansion to this study, as well as coaching teachers and childcare professionals how to implement the intervention at preschool, daycare, and other community-based settings. This research could also be expanded by instructing small groups of parents or by utilizing a group research design. Understanding that constructive play often lays the foundation for imaginary, associative, and cooperative play, future research could explore this influence for children with or at risk for disabilities. Today, many young children demonstrate behaviors due to toxic stress and difficulties with regulating their emotions. Exploring the relationship between constructive play and emotional regulation would be a very useful area for future study. Additionally, although constructive play is most frequently experienced in early childhood, future research is needed to explore how creative constructive play is enjoyed across the lifespan as an adolescent and adult leisure activity.


**Implications for Practice**


The results of this study highlight the importance of empowering parents and provide direction for parents to encourage engaged constructive play in their homes. Parents promoted their child’s hands-on constructive play engagement by addressing their home temporal environment by setting aside time each day for play. They reduced barriers and provided support in the home’s physical/sensory environment by turning off visual distractions such as TV and tablet screens and providing their child choices in play materials. Parents provided social–emotional support by encouraging play to be child-led, being emotionally responsive to their child’s needs, and asking open-ended questions to facilitate their child’s creativity and sharing about what they are creating. Education and professional development for preservice teachers, inservice teachers, and other direct service providers working with young children would expand this naturalistic environmental approach to preschool and other childcare settings. 

## 5. Conclusions

This study confirms that this culturally responsive parent-implemented play intervention increases young children’s constructive play engagement in the home and is meaningful in the daily lives of families. These results establish initial evidence of a functional relation between parent-implemented environmental changes and increased engagement in constructive play, a skill that contributes to child development, school-readiness, and well-being. Findings also suggest that when parents are empowered and understand the importance of play, they prioritize creative constructive play for their children in their home life. It is hoped that these finding can be utilized to expand research and to support changes in policy and practice that emphasize the importance of centering and empowering parents in children’s play, especially for children with or at risk for developmental disabilities. 

## Figures and Tables

**Figure 1 behavsci-14-00247-f001:**
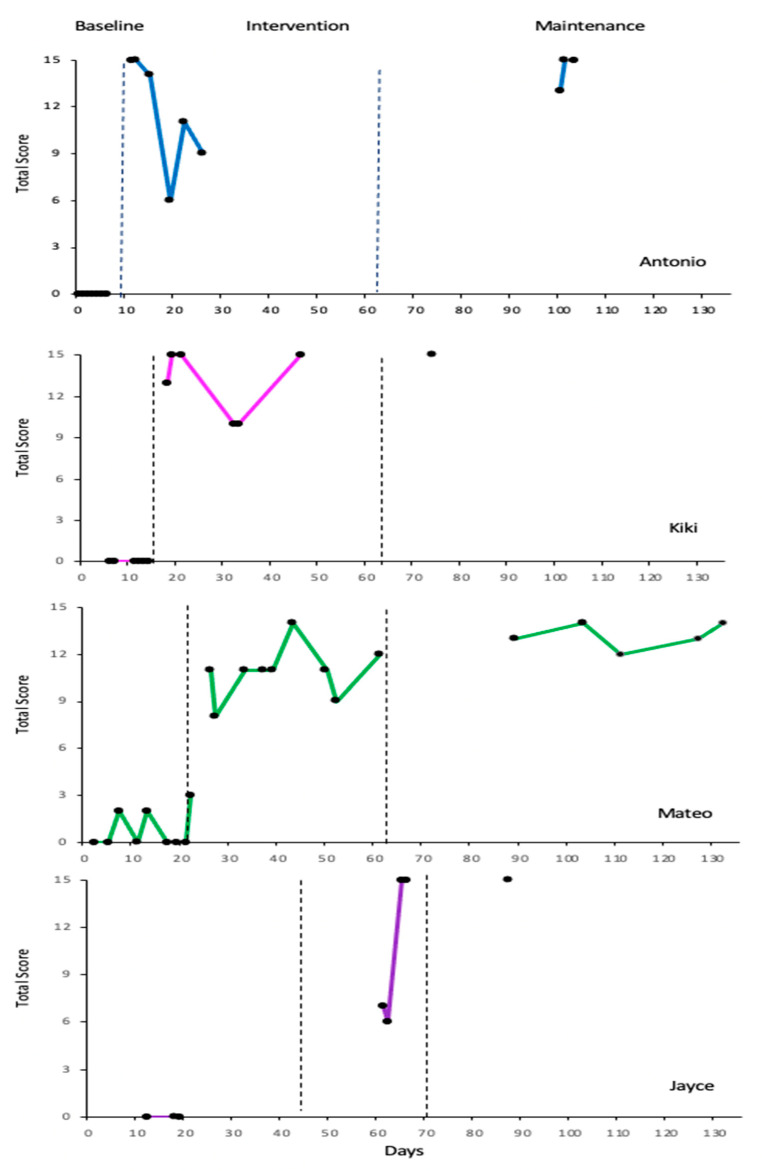
Child’s active engagement in constructive play.

**Table 1 behavsci-14-00247-t001:** Within condition analysis.

Within Condition Measure	Antonio ^a^	Kiki	Mateo	Jayce
Level Mean				
Baseline phase	0	0	0.89	0
Intervention phase	11.33	13	10.89	10.75
Maintenance phase	14.33	15	13.2	15
Level Median				
Baseline phase	0	0	0	0
Intervention phase	12.5	14	11	11
Maintenance phase	15	15	13	15
Level Absolute Change				
Baseline phase	0 − 0 = 0 stable	0 − 0 = 0 stable	3 − 0 = 3 improving	0 − 0 = 0 stable
Intervention phase	15 – 9 = 6 deteriorating	15 − 13 = 2 improving	12 − 11 = 1 improving	15 − 7 = 8 improving
Maintenance phase	15 – 13 = 2 improving	n/a	14 – 13 = 1 improving	n/a
Level Relative Change				
Baseline phase	0 − 0 = 0 stable	0 − 0 = 0 stable	0 − 0 = 0 stable	0 − 0 = 0 stable
Intervention phase	15 − 9 = 6	15 − 10 = 5	11.5 – 11 = 0.5	15 − 6.5 = 9.1
Maintenance phase	15 – 13 = 2 improving	n/a	13.5 − 13.5 = 0 stable	n/a
Level Range				
Baseline phase		0–0	0–3	0–0
Intervention phase	9–15	10–15	8–14	6–15
Maintenance phase	13–25	15	12–14	15
Level Stability				
Baseline phase	100% stable	100% stable	100% stable	100% stable
Intervention phase	66% variable	66% variable	89% stable	0% variable
Maintenance phase	100% stable	n/a	100% stable	n/a
Trend Strength and Direction				
Baseline phase r^2^ = (0.00–1.00)	0.0 zerocelerating	0.0 zerocelerating	0.0461 accelerating	0.0 zerocelerating
Intervention phase r^2^ = (0.00–1.00)	0.5478 decelerating	0.0296 decelerating	0.0828 accelerating	0.9074 accelerating
Maintenance phase r^2^ = (0.00–1.00)	0.75 accelerating	n/a	0.0389 accelerating	n/a
Trend Stability				
Intervention phase stability envelope	10–15	11.2–16.8	8.8–13.2	8.8–13.2
Intervention phase percent of data points within stability envelope (20% above and below median)	4/6 = 66% unstable	4/6 = 66% unstable	8/9 = 89% stable	0/4 = 0% unstable
Maintenance phase range	6.5–15	15	12–14	15
Maintenance phase percent of data points within stability envelope (20% above and below median)	3/3 = 100% stable	n/a ^b^	5/5 = 100% stable	n/a ^b^
Direction of the first 3 data points of the intervention phase used to determine readiness for the next participant to move from baseline to intervention phase	0.058 accelerating	0.571 accelerating	0.563 accelerating	0.886 accelerating

*Notes.* ^a^ All names are pseudonyms to protect the confidentiality of the participants. ^b^ Only one data point so unable to calculate a trend.

**Table 2 behavsci-14-00247-t002:** Between condition analysis.

Between Condition Measure	Antonio ^a^	Kiki	Mateo	Jayce
Range of baseline phase data points	0	0	0 to 3	0
Range of intervention phase data points	6 to 15	10 to 15	8 to 14	6 to 15
Range of maintenance phase data points	13 to 15	15	12 to 14	15
Percentage of nonoverlapping data (PND) between baseline and intervention phases	100%	100%	100%	100%
Percentage of nonoverlapping data (PND) between intervention and maintenance phases	0%	0%	0%	0%
Percentage of overlapping data (POD) between baseline and intervention phases	0%	0%	0%	0%
Percentage of overlapping data (POD) between intervention and maintenance phases	100%	100%	100%	100%
Mean level change between baseline and intervention phases	11.66 Improving	13 Improving	10.013 Improving	10.75 Improving
Mean level change between intervention and maintenance phases	2.67 Improving	2 Improving	2.31 Improving	4.25 Improving
Median level change between baseline and intervention phases	12.5 Improving	14 Improving	11 Improving	11 Improving
Median level change between intervention and maintenance phases	2.5 Improving	1 Improving	2 Improving	4 Improving
Absolute level change between baseline and intervention	15 Improving	13 Improving	8 Improving	7 Improving
Absolute level change between intervention and maintenance phases	4 Improving	0 Stable	1 Improving	0 Stable
Relative level change between the median of the 2nd half of the baseline and the median of the 1st half of intervention phases	14 Improving	15 Improving	10 Improving	6.5 Improving
Relative level change between the median of 2nd half of the intervention and median of the first half of maintenance phases	5 Improving	1 Improving	2 Improving	4 Improving
Trend direction change (Intervention/Baseline)	Decelerating/ Zerocelerating	Decelerating/ Zerocelerating	Accelerating/ Accelerating	Accelerating/ Zerocelerating
Trend direction change (Maintenance/Intervention)	Accelerating/ Decelerating	n/a ^b^ Decelerating	Accelerating/ Accelerating	n/a ^b^ Accelerating
Trend effect on dependent variable	Improving	Improving	Improving	Improving

*Notes.* ^a^ All names are pseudonyms to protect the confidentiality of the participants. ^b^ Only one data point so unable to calculate a trend.

## Data Availability

The raw data supporting the findings of this article are not available due to IRB restrictions.

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
