# Peer review of "Empowering and Educating Parents to Implement a Home Intervention: Effects on Preschool Children’s Engagement in Hands-on Constructive Play"

_behavsci, 2024, doi:10.3390/bs14030247_

Round 1

Reviewer 1 Report

Comments and Suggestions for Authors

This study offers a very comprehensive methodology to approach the study of family play. It is very well grounded and well written. Relevant aspects of validity and reliability are adequately addressed.

I have two concers regarding this paper. First, parents are not described in the participants section, although they are key participants. Second, this paper derives from the first author's doctoral dissertation, yet I could not find this information in the manuscritpt, I found out throught an Internet search. I believe that not stating the provenance of the data and research constitutes self-plagiarism.

Author Response

Please see the attached document for our point-by-point response to reviewer 1's comments.

Reviewer 2 Report

Comments and Suggestions for Authors

A clear articulation of a well designed study. I have just the following suggestions:

1.      The introduction seems to be solely a justification for play using some seminal references – I would usually expect an introduction to do more that, and include recent references to provide an up to date warrant for the research.

2.      A greater focus on constructive play in the literature review section would be useful in providing the warrant for why this was the focus in the study. Constructive play is discussed briefly in the school readiness section - which suggests that school readiness is the reason for doing constructive play. It would be good to articulate the other range of benefits constructive play can provide.

3.      There appears to be a significant number of older references throughout – I wonder if a more recent literature review has been conducted?

Page 3, line 113-120

Clarity is needed in this section. You interchange ‘environmental practices’ with ‘ecological practices’ without defining either. Use of the term practices is a little confusing as the examples include space, family, materials which don’t fit with my understanding of practices – hence the need for a definition of what you mean by environmental/ecological practices. More detailed examples could help.

Page 3, line 127-129

This sentence negates the strengths and skills parents already have and suggests they can only be adept with support from professionals. I think it just needs a reword.

Page 4, line 156-157

Constructive play is not my area of research, however I find it hard to believe there is very limited research in this area. I could be wrong but think about whether ‘very’ is needed and accurate.

Page 17-18

There are a few references to ‘I’ and ‘me’ – wondering if these should re ‘we’ and ‘us’?

Page 19, line 774

Maybe a stretch to suggest these findings confirm the importance of constructive play – what they confirm is the importance of parental planning for, and involvement in, play.

Author Response

Please see the attached document for our point-by-point response to reviewer 2's comments. 
